# 🦜 Lory: Fully Differentiable Mixture-of-Experts for Autoregressive Language Model Pre-training

**Zexuan Zhong[†], Mengzhou Xia[†], Danqi Chen[†], Mike Lewis[‡]**
[†]Princeton University, [‡]Meta AI
{zzhong,mengzhou,danqic}@cs.princeton.edu,mikelewis@meta.com

## Abstract

Mixture-of-experts (MoE) models facilitate efficient scaling; however, training the routing network introduces the challenge of optimizing a non-differentiable, discrete objective. Recently, a fully differentiable MoE architecture called SMEAR was proposed (Muqeeth et al., 2023), which softly merges experts in the parameter space. Nevertheless, its effectiveness has only been demonstrated in downstream fine-tuning on classification tasks. In this work, we present Lory, a novel approach that scales such architectures to autoregressive language model pre-training. Lory introduces two key techniques: (1) a causal segment routing strategy that achieves high efficiency for expert merging operations while preserving the autoregressive nature of language models, and (2) a similarity-based data batching method that encourages expert specialization by grouping similar documents in training instances. We pre-train a series of Lory models from scratch on 150B tokens, with up to 32 experts and 30B (1.5B active) parameters. Experimental results show significant performance gains over parameter-matched dense models in both perplexity (+13.9%) and a variety of downstream tasks (+1.5%-11.1%). Despite segment-level routing, Lory models achieve competitive performance compared to state-of-the-art MoE models with token-level routing. We further demonstrate that the trained experts capture domain-level specialization without supervision. Our work highlights the potential of fully differentiable MoE architectures for language model pre-training and advocates for future research in this area.

## 1 Introduction

Mixture-of-experts (MoE) architectures with sparse activation enable the scaling of model sizes while maintaining high training and inference efficiency (Lepikhin et al., 2021; Fedus et al., 2022; Du et al., 2022; Zoph et al., 2022; Lewis et al., 2021; Zhou et al., 2022; Jiang et al., 2024; Xue et al., 2024; Shen et al., 2024). However, training the routing network (or router) in MoE architectures introduces the challenge of optimizing a non-differentiable, discrete objective (Shazeer et al., 2017; Zoph et al., 2022). Various techniques, such as switch routing (Fedus et al., 2022), top-$k$ expert-choice routing (Zhou et al., 2022), and linear programming (Lewis et al., 2021), have been developed to address this challenge, often requiring carefully designed load-balancing objectives (Fedus et al., 2022) or introducing additional complexity in assignment algorithms (Lewis et al., 2021; Roller et al., 2021).

Recent research is exploring fully differentiable MoE architectures to overcome training challenges. SMEAR (Muqeeth et al., 2023), for instance, softly merges experts by averaging their parameters rather than activating the top-$k$ experts. However, SMEAR's effectiveness has only been shown in small-scale fine-tuning experiments on downstream classification tasks (Wang et al., 2018).

In this work, we propose Lory[1], the first approach that scales such fully differentiable MoE architectures to autoregressive *language model pre-training*. Unlike text classification tasks,

---

[1]Lory is a tribe of parrots known for their rainbow-like colors, reflecting the spirit of 'soft' MoE.

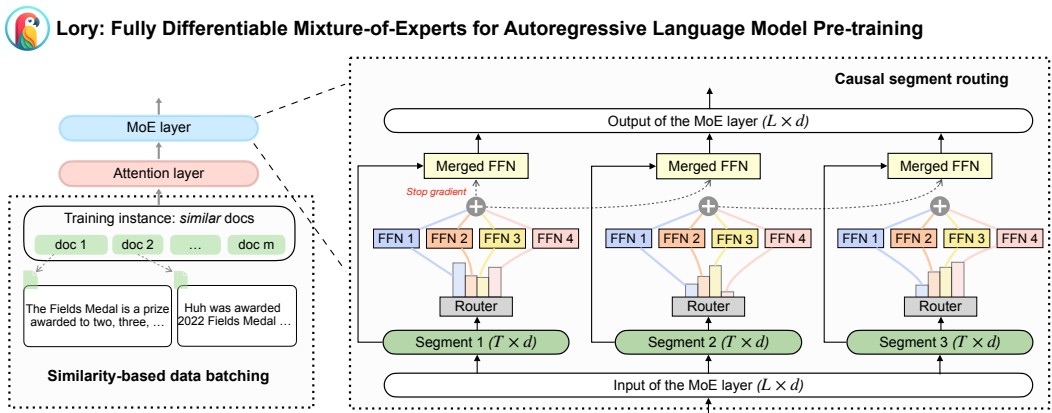

**Figure 1:** We propose 🦜 Lory, a fully differentiable MoE architecture designed for autoregressive language models based on expert merging (Section 2.2). We introduce two key techniques to train Lory: First, we propose the *causal segment routing* strategy, which conducts expert merging at the segment level and preserves the autoregressive property of language models. Second, we use the *similarity-based data batching* method to construct training instances, which steers the experts toward specializing in specific domains or topics.

which only require routing each input sequence to different experts, language modeling makes predictions for each input token. Performing token-level routing is prohibitively expensive because the computational cost of merging operations scales linearly with the number of experts.

Lory is based on two key techniques (Figure 1). First, we propose *causal segment routing*: for a sequence of input tokens, we split them into multiple segments of fixed length and use the previous segment to determine the router's weights and calculate the merged expert for the subsequent segment. During inference, we can simply use the prompt to make a single routing decision throughout the generation process. This segment-level routing strategy preserves the autoregressive nature of language models while keeping the merging operations efficient. However, since the text data for pre-training language models usually concatenates random sets of documents, we find that such routing can lead to scenarios in which experts are not sufficiently specialized. To address this, we propose our second technique: *similarity-based data batching* for MoE training, which groups semantically similar documents to form consecutive segments. This idea has been recently proposed to train LMs to better reason across document boundaries (Shi et al., 2024), and we find that it leads to more effective training of expert routing.

We pre-train a series of Lory models from scratch under a training budget of 150B tokens, with 0.3B and 1.5B active parameters, and 8, 16, or 32 experts (corresponding to 6.8B and 29.5B full parameters at most; see Table 4). Experimental results show that our Lory models significantly outperform equal-sized dense models trained with the same amount of data, achieving performance gains in both perplexity (+13.9%) and a wide range of downstream tasks, including commonsense reasoning (+3.7%), reading comprehension (+3.3%), closed-book QA (+1.5%), and text classification (+11.1%). Interestingly, despite using segment-level routing, Lory achieves competitive performance compared to state-of-the-art MoE models with *token-level*, non-differentiable discrete routing (Zhou et al., 2022). Our analysis further shows that the trained experts capture domain-level specialization without any supervision, making it distinct from previous MoE LMs with token-level routing, which only exhibit local patterns uniformly distributed across different domains (Xue et al., 2024; Jiang et al., 2024). Together, we present the first fully differentiable MoE model suitable for language model pre-training and demonstrate its effectiveness at scale. We hope our work sheds light on the potential of fully differentiable MoE architectures in cultivating specialized experts and encourages continued exploration in this research field.

## 2 Preliminaries

### 2.1 Sparsely-activated MoE

Transformer-based MoE language models typically substitute feed-forward network (FFN) layers with sparsely activated MoE layers (Shazeer et al., 2017; Fedus et al., 2022; Zoph et al., 2022). Assume an MoE layer consists of $E$ expert FFNs, each parameterized as $\text{FFN}(\cdot; \theta_1), \ldots, \text{FFN}(\cdot; \theta_E)$, where the function $\text{FFN} : \mathbb{R}^d \to \mathbb{R}^d$ defines a single expert module. For each token $x$ in a sequence, an MoE layer takes the hidden representation $h_x \in \mathbb{R}^d$ as the input and computes its output $o_x \in \mathbb{R}^d$ by sparsely activating $k$ experts in this layer and aggregating the outputs through a weighted sum:

$$o_x = \sum_{i=1}^{E} e_i \cdot \text{FFN}(h_x; \theta_i), \quad \text{where} \quad e_i = \text{Top-}k(\text{Softmax}(R(h_x)))_i. \tag{1}$$

The routing weight $e_i$ for the $i$-th expert is determined by a *routing network* or *router $R$*, which takes $h_x$ as input and calculates the weight for each expert. In practice, to achieve sparsity and computational efficiency, only one (Fedus et al., 2022) or $k$ (Lepikhin et al., 2021) experts with the highest routing weights are activated at each MoE layer. The weights of the remaining experts are set to 0 (i.e., $e_i = 0$), eliminating the need to compute $\text{FFN}(h_x; \theta_i)$ and effectively deactivating the $i$-th expert.

### 2.2 Fully Differentiable MoE Architectures via Expert Merging

The primary challenges in training sparsely activated MoE models arise from the difficulty in training discrete routers. A promising direction is to design fully differentiable MoE architectures that do not rely on extra loss formulations for stabilized training. A recent model architecture (Muqeeth et al., 2023) demonstrates the feasibility of this approach by computing a weighted average of all expert FFNs in the parameter space (Matena & Raffel, 2022; Wortsman et al., 2022), thereby creating a "merged FFN." Given an input $x$ and its corresponding routing weights $e_i$, the output $o_x$ of a merged FFN is computed as:

$$o_x = \text{FFN}(h_x; \sum_{i=1}^{E} e_i \cdot \theta_i), \quad \text{where} \quad e_i = \text{Softmax}(R(h_x))_i. \tag{2}$$

However, naively extending this approach to autoregressive language models, which would require computing the merged FFN for each token in a sequence, is infeasible as the computational costs of merging operations scale linearly with the number of experts. SMEAR (Muqeeth et al., 2023) has only been evaluated for downstream fine-tuning on text classification tasks, making routing decisions based on a pooling representation of the entire input sequence, i.e., $e_i = \text{Softmax}(R(\frac{\sum_{j=1}^{L} h_{x_j}}{L}))_i$. Such operations disrupt the autoregressive property in language model pre-training. In this work, we address these challenges by developing a fully differentiable MoE architecture suitable for autoregressive language modeling and pre-training such models at scale.

## 3 Our Approach: 🦜 Lory

Lory is an approach for pre-training fully differentiable MoE language models (Figure 1). The core technique that enables Lory to be fully differentiable is expert merging (Muqeeth et al., 2023, see details in Section 2.2). To make it computationally feasible, we propose a causal segment routing method that merges experts only once per segment, effectively reducing the number of merging operations (Section 3.1). We also propose a data batching strategy that groups semantically similar texts, which is crucial for the effective training of the segment-level router (Section 3.2).

**Notations.** We denote an input sequence of $L$ tokens as $X = (x_1, x_2, \ldots, x_L)$. Considering a segment size $T$, we divide the input sequence into $N = \lceil L/T \rceil$ segments, denoted as

$S_1, S_2, \ldots, S_N$. We use $R$ to denote the routing network (parameterized as a linear layer), which computes the weights for expert merging. Let $h_x$ represent the hidden representation of the token $x$. The parameters of the $i$-th expert FFN are denoted by $\theta_i$.

## 3.1 Efficient Expert Merging via Causal Segment Routing

**Challenges.** An intuitive way to reduce the computational cost is to use segment-level routing instead of token-level routing, which can reduce the number of merging operations from $L$ to $N$. However, simply using the current segment to compute the routing weights can cause information leakage.

**Training design.** We propose *causal segment routing* to effectively route information across segments in an autoregressive manner.[2] This method merges FFNs in an MoE layer based on the previous segment's information and uses it to process the current segment. Specifically, given a training instance $X$ that consists of $L$ tokens (e.g., $L = 4096$), we split the training instance into $N$ segments, each containing $T$ (e.g., $T = 256$) consecutive tokens. For the $k$-th segment $S_k$, where $k > 1$, we compute the average of the hidden representations of its preceding segment $S_{k-1}$, denoted as $\bar{h}_{k-1}$. Using the average hidden representation allows the model to adapt to prompts of varying lengths during inference. $\bar{h}_{k-1}$ is then utilized to determine the routing weights, resulting in a merged expert $\bar{\theta}$:

$$\bar{h}_{k-1} = \frac{1}{T} \sum_{x \in S_{k-1}} h_x, \quad e_i = \text{Softmax}(R(\bar{h}_{k-1})), \quad \bar{\theta} = \sum_i e_i \cdot \theta_i. \tag{3}$$

We then use the merged expert $\bar{\theta}$ to process all the tokens in the current segment $S_k$, i.e., $o_x = \text{FFN}(h_x; \bar{\theta}), \forall x \in S_k$. This approach ensures that the routing decisions made by the model are based exclusively on data from preceding positions. For the first segment $S_1$, the representation of the segment itself is used to compute the merging weights for its own FFN. To prevent information leakage, we implement a stop-gradient operation on $R(\bar{h}_1)$. As demonstrated in Appendix B, merging experts at the segment level incurs minimal overhead compared to the training of dense models.

**Prompt-only routing during inference.** During inference, we begin with a given prompt and make a single routing decision per layer based on the average hidden representations of the prompt. This routing decision determines a merged FFN, which is used consistently throughout the entire generation process. It is important to note that this inference process is as simple and computationally efficient as that of dense models.[3]

## 3.2 Similarity-based Data Batching

The standard practice of pre-training LMs is to randomly concatenate documents to construct training instances with a fixed length. This approach can lead to under-specialized experts because tokens within adjacent segments may come from very different and irrelevant documents.

To mitigate this issue, we employ a similarity-based data batching technique inspired by Shi et al. (2024), which sequentially concatenates similar documents to construct training instances. This method encourages high similarity between adjacent segments, enabling the experts to specialize in different domains or topics.

We use Contriever (Izacard et al., 2022) to measure document similarity and apply a greedy search algorithm to concatenate similar documents to form batches (see Appendix C).

---

[2]Pseudocode of the *causal segment routing* strategy can be found in Appendix A.

[3]In Appendix G.2, we compare the prompt-only routing strategy to using the causal segment routing strategy that faithfully follows the training design, and find they do not lead to significant differences. We also discuss the potential of converting Lory to sparsely MoE models for memory-efficient inference.

Although our data batching technique is similar to that of Shi et al. (2024), our goal is different. While they focus on enhancing language models' reasoning across document boundaries, we find this approach particularly effective for fostering expert specialization during MoE model training.

## 4 Experiments

In this section, we evaluate Lory by training a series of language models from scratch. We first describe the experimental setups (Section 4.1) and then present the results (Section 4.2).

### 4.1 Setups

**Models.** We evaluate our approach by training decoder-only Transformer models which consist of 0.3B and 1.5B active parameters.[4] For each FFN layer in the Transformer model, we replace it with MoE layers with $E \in \{8, 16, 32\}$ experts with exactly the same architecture.[5] Appendix D shows the configuration of model architectures as well as the total parameter count. We follow LLaMA (Touvron et al., 2023a) and use SwiGLU (Shazeer, 2020) as the activation function in FFNs. We use the same tokenizer as the LLaMA models (Touvron et al., 2023a;b). All models are trained with a 4096-token context window. In the causal segment routing strategy, we set the length of each segment to be $T = 256$.

**Training details.** We employ the AdamW optimizer (Loshchilov & Hutter, 2019) with $\beta_1 = 0.9$ and $\beta_2 = 0.95$ and use a learning rate of 2e-4 with a cosine learning rate scheduler. All models with a batch size of 1 million tokens. We employ the data parallelism with the ZeRO optimization (Rajbhandari et al., 2020) for distributed training. At the beginning of training, we train a parameter-matched dense model and duplicate the FFN layers as initialization of the MoE model. In our experiments, we use the first 5% training steps as the warmup to initialize the MoE weights. We find that without warmup training, there may be more experts under-utilized (see Appendix G.3 for an ablation study). We also apply a linear warmup to the learning rate scheduler for the first 5% training steps. We train our models with up to 64 A100 GPUs.

**Training datasets.** We randomly sample a subset (150 billion tokens) of the Commoncrawl dataset (Wenzek et al., 2019) for training. Using the similarity-based data batching method from Shi et al. (2024), we construct all training instances (see Appendix C for details).

**Evaluation datasets.** We evaluate all the models on language modeling tasks by measuring the perplexity of trained models on held-out evaluation datasets sampled from arXiv, Books, Wikipedia, C4 (Raffel et al., 2020), and Python code (a Python subset of Github). Each evaluation dataset contains 1K samples, each of which consists of 4096 tokens.

We also evaluate models in downstream tasks with in-context learning (Brown et al., 2020), including common sense reasoning: BoolQ (Clark et al., 2019), PIQA (Bisk et al., 2020), SIQA (Sap et al., 2019), HellaSwag (Zellers et al., 2019), WinoGrand (Sakaguchi et al., 2020); reading comprehension: RACE (Lai et al., 2017), ARC (Clark et al., 2018)); closed-book QA: Natural Questions (Kwiatkowski et al., 2019), TriviaQA (Joshi et al., 2017); and text classification: AGNews (Zhang et al., 2015), SST-2 Socher et al. (2013), Amazon and Yelp (Zhang et al., 2015), FEVER (Thorne et al., 2018), MRPC (Dolan & Brockett, 2005). For text classification tasks, we follow the evaluation setup of Min et al. (2022); for the rest of tasks, we follow the same setup as Touvron et al. (2023b).

---

[4]Here, "active parameters" refers to the size of the model after merging at each MoE layer.

[5]In Appendix E, we additionally conduct experiments on a 7B dense model and a 7B/4E MoE model *without* using similarity-based data batching. Due to the limited computing resources, we are not able to train 7B models on the similarity-based batched dataset.

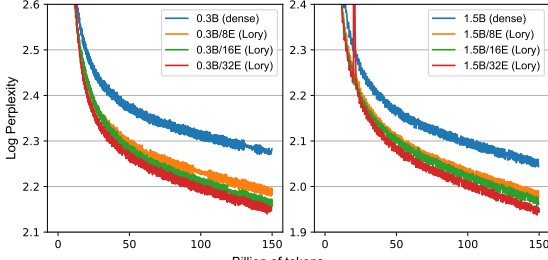

| Model | arXiv | Books | Wiki | C4 | Python |
|---|---|---|---|---|---|
| 0.3B | 8.4 | 18.0 | 10.3 | 13.8 | 15.2 |
| 0.3B/8E | 7.4 | 16.0 | 9.2 | 13.3 | 12.5 |
| 0.3B/16E | **7.2** | 15.7 | 9.1 | 13.1 | 12.2 |
| 0.3B/32E | **7.2** | **15.5** | **8.9** | **13.0** | **11.7** |
| 1.5B | 6.6 | 13.6 | 7.8 | 10.7 | 10.4 |
| 1.5B/8E | 6.2 | 12.8 | 7.6 | 10.6 | 10.1 |
| 1.5B/16E | 6.0 | 12.4 | **7.1** | 10.6 | 8.9 |
| 1.5B/32E | **5.8** | **12.3** | **7.1** | **10.4** | **8.7** |

**Figure 2:** Left: training curves (log perplexity) of models with different sizes and experts. Right: Perplexity of trained models on different evaluation sets (arXiv, Books, Wikipedia, C4, and Python). We include the detailed model configurations and sizes in Appendix D.

| | Commonsense Reasoning | | | | | Reading Comprehension | | | |
|---|---|---|---|---|---|---|---|---|---|
| Model | PIQA | SIQA | BoolQ | HellaSwag | WinoGrande | RACE-m | RACE-h | ARC-e | ARC-c |
| 0.3B | 65.8 | 42.7 | 44.6 | 34.6 | 51.2 | 41.7 | 30.9 | 51.5 | 21.3 |
| 0.3B/8E | 67.5 | 41.2 | 41.2 | 34.8 | **54.4** | 43.1 | 31.4 | 52.4 | 22.1 |
| 0.3B/16E | 67.2 | **44.1** | 56.6 | **34.9** | 54.1 | **43.9** | 31.1 | 54.8 | 24.9 |
| 0.3B/32E | **68.2** | 43.0 | **58.0** | 34.7 | 53.4 | 42.7 | **32.0** | **57.4** | **26.3** |
| 1.5B | 71.2 | 45.0 | 54.0 | **43.9** | 60.9 | 50.1 | 36.7 | 65.0 | 31.0 |
| 1.5B/8E | **72.1** | 45.2 | **62.0** | 43.6 | **63.7** | 51.2 | 36.5 | 66.3 | 32.5 |
| 1.5B/16E | 71.3 | 45.0 | 56.0 | 43.7 | 61.5 | **51.7** | **37.3** | 66.3 | **32.7** |
| 1.5B/32E | **72.1** | **47.1** | 59.9 | 43.8 | 61.9 | 51.5 | 32.4 | **66.7** | **32.7** |

| | Closed-book QA | | Text Classification | | | | | | Avg |
|---|---|---|---|---|---|---|---|---|---|
| Model | NQ | TQA | AGNews | Amazon | SST-2 | Yelp | Fever | MRPC | |
| 0.3B | 4.7 | 8.8 | 30.3 | 53.6 | 54.6 | 66.0 | 47.6 | 62.0 | 41.8 |
| 0.3B/8E | 5.3 | 9.0 | 38.4 | 52.3 | 54.6 | 62.6 | 56.6 | 59.0 | 42.7 |
| 0.3B/16E | **6.0** | 10.2 | 36.3 | **75.6** | 53.3 | 64.0 | **57.0** | **65.0** | 45.8 |
| 0.3B/32E | 5.3 | **10.2** | **47.3** | 64.0 | **55.3** | 73.3 | 55.7 | 56.0 | **46.0** |
| 1.5B | **7.6** | 23.8 | 64.0 | 65.3 | 80.0 | 58.6 | **59.0** | 66.7 | 51.9 |
| 1.5B/8E | 7.3 | 24.2 | **65.0** | 94.0 | 80.0 | 88.3 | 57.0 | 64.0 | 56.1 |
| 1.5B/16E | 7.3 | **25.6** | 61.6 | 78.3 | 84.6 | 93.6 | 57.3 | 63.6 | 55.1 |
| 1.5B/32E | 7.0 | 25.4 | 62.3 | **94.7** | **85.0** | **95.3** | 56.3 | **66.7** | **56.5** |

**Table 1:** We compare the Lory MoE models with the parameter-matched dense models on downstream tasks, including commonsense reasoning, reading comprehension, closed-book QA, and text classification.

## 4.2 Main Results

**Training efficiency and convergence.** Figure 2 (left) shows the training loss curves of the dense model and our MoE models with different model sizes. First, we find that with the same amount of training tokens, our models clearly achieve better training loss compared to the dense model baseline. For the 0.3B and 1.5B models, our models with 32 experts achieve the same level of loss with fewer than half of the training tokens. This indicates that our approach achieves much better performance with the same training compute (see analysis of additional FLOPs from MoE layers in Appendix B). We also observe that when using more experts, we are able to gain more improvement.

**Language modeling.** We evaluated the models on language modeling tasks and found that our MoE models consistently outperformed the dense baseline, significantly reducing perplexity across all domains. For instance, the 0.3B/32E model improved perplexity by 13.9% on Books compared to the 0.3B dense model. Notably, improvements were most pronounced in test domains distinct from the training data (e.g., Python), indicating strong expert specialization, which is further explored in Section 5.4).

**Downstream tasks.** Table 1 shows the model performance on downstream tasks. We observe significant performance across all tasks. For example, our 0.3B/32E model achieves an

average performance improvement of +3.7% in common sense reasoning, +3.3% in reading comprehension, +1.5% in reading comprehension, and +11.1% in text classification.

## 5    Analysis and Ablation Studies

In this section, we conduct ablation studies and analysis to understand the significance of each component of our approach. In Appendix G, we provide additional insights, demonstrating that (1) during the inference of downstream tasks, whether the entire input prompt is routed once or each segment is routed individually does not result in substantial differences in the tasks we evaluated; and (2) warmup training is crucial for achieving high expert utilization, particularly when training MoE models with a large number of experts.

### 5.1    Importance of Causal Segment Routing

We compare our causal segment routing strategy with an alternative *prefix routing* strategy for training. In prefix routing, expert merging is performed only once for each sequence based on the first segment. The merged FFN is then used to process the rest of the sequence without further updates. Figure 3 shows that using only a prefix for routing leads to much worse performance compared to causal segment routing. These results highlight the importance of using every segment to provide strong training signals for routers.

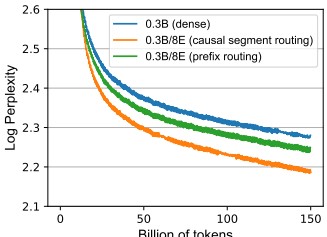
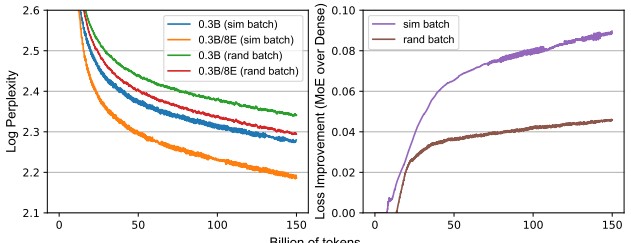

**Figure 3:** Training curves of causal segment routing and prefix routing. The latter is a straightforward segment-level routing strategy that uses the first segment to route the entire input.

**Figure 4:** Left: Training curves of similarity-based data batching (*sim batch*) or the standard random batching (*rand batch*). Right: Training loss difference between Lory and a dense model when using different batching strategies. Lory leads to a larger loss improvement over the dense model when using similarity-based data batching.

### 5.2    Importance of Similarity-based Data Batching

To investigate the importance of similarity-based data batching, we compare the performance improvement of MoE models over dense models with and without this batching method. Figure 4 (left) shows the training loss of dense (0.3B) and MoE models with eight experts (0.3B/8E) using similarity-batched (sim batch) and randomly-batched (rand batch) data. MoE models consistently outperform dense models in both setups. However, the loss improvement (i.e., the difference in loss between dense and MoE models) is much larger with similarity-based batching, and this effect is amplified with more training data (Figure 4 (right)). These results strongly support the importance of similarity-based batching for effectively training our MoE model.

### 5.3    Comparison to Existing MoE Models

We compare our approach with Expert Choice (EC) (Zhou et al., 2022), a state-of-the-art MoE method that ensures balanced load during training by having each expert select top-$k$ inputs according to the routing weights. During inference, we route each token into top-$k$ experts to avoid to leverage global information for routing.

We consider two variants of EC MoE models, both with a capacity factor of 1 to match the computation of our MoE models. First, we train a sparse EC MoE model using our segment routing strategy, where each expert selects top segments and processes all tokens within those segments. This variant allows us to directly compare our expert-merging strategy with the expert choice method while using the same segment-level routing approach. Second, we consider the original EC setting with token-level routing to provide an end-to-end comparison with state-of-the-art MoE models using the same amount of training computation. Figure 5 shows the training loss curves. We observe that Lory (blue curve) significantly outperforms segment-level EC (orange curve) with the same routing setting, suggesting that a fully differentiable architecture is more effective than a sparse MoE when using the same routing

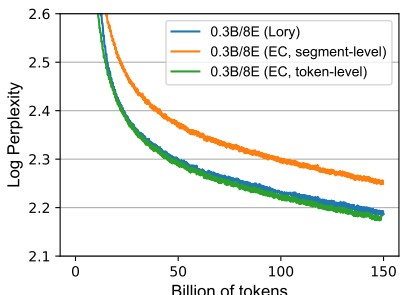

**Figure 5:** Comparison with the state-of-the-art MoE training technique Expert Choice (EC) with a segment-level or token-level routing. For both EC models, we use the capacity factor of 1 with the same amount of FLOPs as our training method for the fair comparison.

strategy. Comparing Lory with the token-level EC model (green curve), we find that Lory achieves competitive results despite using segment-level routing and not requiring any advanced training techniques. These results highlight the significant potential of Lory.

Table 2 shows model perplexity on held-out evaluation sets. The token-level EC model outperforms ours on C4, likely due to its similarity to the training set (Commoncrawl). However, on arXiv, Books, and Wikipedia, EC performs similarly or slightly worse. Notably, our model excels on the Python evaluation (12.5 vs. 13.6 perplexity), suggesting segment-level routing can be particularly effective for out-of-domain data (i.e., Python in CommonCrawl). Our analysis in Section 5.4 shows that segment-level routing models are indeed able to learn experts that are specialized in specific domains (e.g., Python code), potentially helping models achieve high performance in less frequent domains.

| Model | arXiv | Books | Wiki | C4 | Python |
|---|---|---|---|---|---|
| 0.3B/8E (Lory) | **7.4** | **16.0** | **9.2** | 13.3 | **12.5** |
| 0.3B/8E (EC, seg-level) | 7.9 | 17.6 | 10.5 | 14.1 | 14.1 |
| 0.3B/8E (EC, tok-level) | 7.5 | 17.0 | **9.2** | **12.8** | 13.6 |

**Table 2:** Perplexity of our trained MoE model and EC models on evaluation sets. We instantiate EC methods with our segment-level routing and the original token-level routing.

## 5.4 Expert Utilization and Specialization

**Utilization: How many experts are actively utilized?** One potential issue of training MoE models is the models may collapse to dense models because most experts are under-utilized (e.g., some experts have never been activated). In Appendix G.1, we show although without using any auxiliary loss on load balancing, Lory is able to achieve high expert utilization, preventing the MoE models from collapsing to dense models.

**Specialization: What do experts learn?** In order to study the expert specialization, we investigate the averaged routing weights at different layers of the 0.3B/8E model, on different domains (Books, arXiv, Python, and Wikipedia). Figure 6 shows the routing weights at layer 0, 11, and 23 (the first, middle, and last layer) of the 0.3B/8E model.[6] First, we find that there exists clear domain-level expert specialization in our trained MoE models, even though no additional domain-level supervision is used during training. For instance, expert 7 at layer 11 is specialized to process inputs in the arXiv domain. We also observe that routing weights on arXiv and Python code are more similar compared to Books and Wikipedia, likely because LaTex code and Python code are dissimilar to natural language.

---

[6]In Appendix F, we show the averaged routing weights at all layers of the 0.3B/8E model.

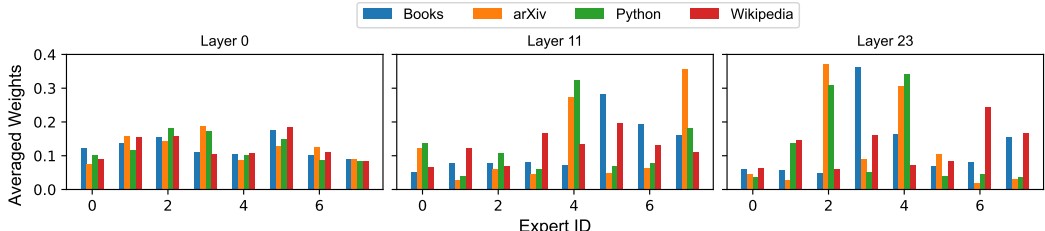

**Figure 6:** Averaged routing weights at layer {0, 11, 23} of the 0.3B/8E model on different domains (Books, arXiv, Python, Wikipedia). We observe that the experts in our MoE models learn domain-level specialization, especially at middle and higher layers.

Second, experts at the middle or high layers are more specialized in specific domains, while the routing weights at lower layers are similar and flat across domains.

It is worth noting that our learned experts behave differently from those of prior token-level MoE models, where shallow token-level specialization is observed. For example, some experts are specialized for a specific type of word (e.g., punctuations, articles), and few deep semantic features are captured by the learned routers (Jiang et al., 2024; Lewis et al., 2021; Zoph et al., 2022; Shazeer et al., 2017; Xue et al., 2024). Our models learn domain-level specialization, which we attribute to the segment-level routing strategy used during training. This strategy allows routers to capture global semantic features beyond the token level. The complementary nature of features captured by segment/sentence-level and token-level routing strategies suggests the possibility of combining them to build even stronger models, and we leave it for future work.

## 6   Discussion of Parallelism Strategies for Scaling Lory

In our experiments, we utilized data parallelism with ZeRO optimization (Rajbhandari et al., 2020) to reduce memory usage. Data parallelism involves either maintaining a copy of the model parameters on each device or using fast communication between devices to synchronize parameters with ZeRO optimization. As we scale up the Mixture of Experts (MoE) models to sizes exceeding 100 billion parameters, the substantial increase in MoE parameters can lead to a bottleneck due to the significant amount of parameter communication required.

To address this problem, expert parallelism is utilized for sparsely MoE models, where experts are distributed across different devices and operate independently (Lepikhin et al., 2021; Fedus et al., 2022). Unlike traditional MoE models, which use a top-k discrete routing network to dispatch inputs to a subset of experts, our soft-routing MoE models require access to all expert parameters for computations.

To overcome this challenge, we can employ an *expert-wise model parallelism* strategy, as shown in Figure 7 (left). This involves partitioning the MoE layers across multiple devices by sharding all experts along with their hidden dimension. Consequently, identical dimensions from different experts are allocated to the same device. We shard all the experts along with the hidden dimension and distribute the same shard of different experts to one device. This expert-wise model parallelism enables us to communicate only hidden activations instead of model parameters, which is more efficient since hidden activations do not scale with the number of experts. Furthermore, for non-MoE components, such as attention layers, we can employ data parallelism to minimize communication overhead. Figure 7 (right) illustrates a training setup in which we use a combination of data parallelism and model parallelism to facilitate training at scale. We plan to explore the experiments for scaling our approach to very large models in future work.

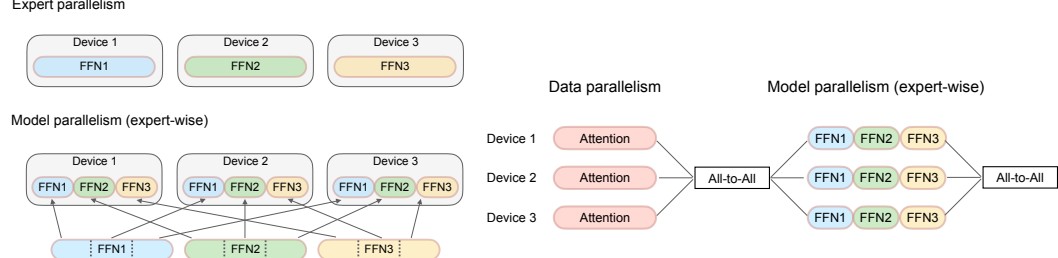

**Figure 7:** Left: Comparison between expert parallelism and expert-wise model parallelism. Right: Using a mixed strategy of data parallelism and expert-wise model parallelism. We use data parallelism for non-MoE components (e.g., attention) and model parallelism for MoE layers.

# 7    Related Work

**Mixture of Experts.**    Sparsely activated MoE models (Shazeer et al., 2017) have been proposed to demonstrate the potential of massively scaling up model sizes. GShard (Lepikhin et al., 2021) adapts the sparse MoE architecture into Transformer models and achieves strong results on machine translation. Recent work has extended it to general language models (Fedus et al., 2022; Zoph et al., 2022; Jiang et al., 2024; Dai et al., 2024; Zhou et al., 2022; Du et al., 2022; Artetxe et al., 2021; Xue et al., 2024). Traditional MoE models are trained to route given inputs to one or a few specialized expert modules, which introduces a non-differentiable, discrete decision-learning problem. These existing models are trained with the top-1 or top-2 routing strategy on a carefully designed load balancing objective (Lepikhin et al., 2021; Fedus et al., 2022; Zoph et al., 2022), or employ complicated assignment algorithms to distribute inputs (Lewis et al., 2021; Roller et al., 2021; Zhou et al., 2022). Training MoE models has been shown to be difficult, facing the issues of training instability, expert under-specialization, poor training efficiency (Zoph et al., 2022).

Our approach enables end-to-end gradient back-propagation using fully differentiable MoE architectures. While SMEAR (Muqeeth et al., 2023) and Soft MoE (Puigcerver et al., 2024) also achieve this, SMEAR is limited to text classification tasks with an encoder backbone, and Soft MoE is only evaluated on vision tasks. Lory is the first to scale this architecture for autoregressive language model pre-training. Extending Soft MoE to decoder language models is left for future work.

**Similarity-based data batching.** There exists research that applies a similar data batching method during training. In-context pre-training (Shi et al., 2024) groups relevant documents together to encourage language models to leverage long-range contexts and improve the results of in-context learning and retrieval augmentation. Zhong et al. (2022) batch documents with high lexical similarity to collect more positive pairs in a contrastive learning framework to provide stronger training signals. Despite sharing a similar idea, the goal of our data batching method is to avoid routing irrelevant documents together, which may hurt the expert specialization.

# 8    Conclusion

In this paper, we introduce Lory, a fully differentiable MoE model specifically designed for autoregressive language model pre-training. Through extensive experiments, we demonstrate that Lory significantly outperforms its dense counterparts in both language modeling and downstream tasks. Our findings reveal that the trained experts in Lory are highly specialized and adept at capturing domain-level information. Future research directions include scaling up Lory further (Section 6), integrating token-level and segment-level routing, and developing efficient decoding methods tailored for Lory.

## Acknowledgements

We appreciate the valuable comments and feedback from members of the Princeton NLP group. We also thank Weijia Shi for the help with the experiments and discussions related to the similarity-based data batching method.

## Ethics Statement

This paper presents a novel approach for training sparse large language models. It is crucial to acknowledge that, similar to existing language models, those trained using our method may have comparable societal consequences. For instance, language models can generate factually inaccurate outputs and unethical content, posing risks of spreading misinformation and perpetuating harmful stereotypes or biases (Zhao et al., 2024). Furthermore, malicious users may extract the training data used to develop these models, leading to potential privacy and licensing issues (Carlini et al., 2021). We recognize these potential negative consequences and advise those employing our approach to remain vigilant about these risks when developing powerful language models.

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

# A   Pseudocode of Causal Segment Routing

We include a pytorch-style pseudocode of our proposed *causal segment routing* strategy in Algorithm 1.

---

**Algorithm 1** Pseudocode of causal segment routing.

---

```
# B: batch size (number of training instances)
# L: length of each training instance
# d: hidden dimension
# E: number of experts
# T: length of each segment
# R: routing network (input: hidden rep, output: routing weights)

input x # x: input tensor (BxLxd)

N = L // T # number of segments in each sample
seg_x = x.view(B*N, T, d) # split x into segments

# representation of each segment (BNxE)
repr = mean(seg_x, dim=1)

# routing results (not causal) (BNxE)
e = softmax(R(repr), dim=-1)

# routing results for the first segment
e_first = e.view(B, N, E)[:, 0]

# make causal routing results (shift 1)
e = roll(e, 1) # shift by 1

# set routing results of the first segment
e = e.view(B, N, E) # back to the instance view
e[:, 0] = stop_grad(e_first) # assign w/ stop gradient
e = e.view(B*N, E)

# MoE FFN forward with expert weights e
seg_y = moe_ffn(seg_x, e) # seg_y: B*N x T x d

# back to the instance view
y = seg.y.view(B, L, d)

return y
```

---

moe_ffn: compute the merged expert and process the input (equation 2).

# B   Computational Overhead of Routing and Merging

Here we investigate the computational overhead of our Moe models compared to the dense counterpart. We consider an MoE layer and an input tensor $x$ consisting of $L$ tokens and $d$ dimensions: $x : L \times d$. We assume that the model uses SwiGLU as the activation function in FFNs and it up-projects the input $x$ to $d'$-dimensional activations in FFNs. In this case, processing the input on an FFN requires roughly $6 \times L \times d \times d'$ FLOPs (there are two up projections and one down projections in SwiGLU-based FFNs). The overhead of soft-routing MoE comes mainly from the merging operation. Suppose that there are $E$ experts and that the model makes a routing decision for every segment of $T$ tokens (equivalently, there are $L/T$ routing decisions). Each merging operation on $E$ experts takes $6 \times E \times d \times d'$ FLOPs (we compute three merged matrices). Therefore, the total overhead will be $\frac{L}{T} \times 6 \times E \times d \times d'$ FLOPs. This indicates that compared to a dense FFN layer, an MoE layer with $E$ experts requires $\frac{E}{T}$ more FLOPs, compared to the dense counterpart. In our experiments, we set $T = 256$; this suggests that using $E = 8$ experts introduces 3.1% more computations and using $E = 32$ experts introduces 12.5% more computations at the FFN/MoE layers. It is worth noting that the computations from FFN layers are only a subset of the full model computations, so 3.1% is an overhead upperbound when measuring on full models. In our experiments, our most straightforward implementation leads to a 15% or 28% slowdown of training efficiency when using 8 or 32 experts (Table 3).

| Model | Throughput (tokens/s/gpu) |
|---|---|
| 0.3B | 29,000 |
| 0.3B/8E | 24,500 |
| 0.3B/16E | 22,900 |
| 0.3B/32E | 20,800 |

**Table 3:** Training throughput (tokens/s/gpu) of our MoE models and the dense counterpart. Our implementation is based on data parallelism with the ZeRO optimization (Rajbhandari et al., 2020).

## C  Details of Similarity-based Data Batching

We adapt the pipeline of in-context pre-training (Shi et al., 2024) in our approach. Given a set of documents $\mathcal{D}$, for each document $d \in \mathcal{D}$, we first use Contriever (Izacard et al., 2022) to retrieve top-$k$ most similar documents $N(d)$. The similarity between the document $d_i$ and $d_j$ is defined as the cosine similarity of their Contriever embeddings, i.e., $\text{sim}(d_i, d_j) = \cos(C(d_i), C(d_j))$, where $C$ denotes the Contriever encoder model. We implement an efficient approximate nearest-neighbors search based on the FAISS library (Johnson et al., 2019). Then, we sort all the documents according to the similarity and construct training instances by batch consecutive documents. We use the same greedy algorithm as Shi et al. (2024). We start from a single document and repeatedly add the document that has the highest similarity value and has not been added to the list; we restart the process with a new document if all documents that are connected to the last document of the list are selected. We repeat this process until there are no documents left.

| Model | $n_{\text{params}}$ | $N$ | $D$ | $n_{\text{head}}$ |
|---|---|---|---|---|
| 0.3B | 0.3B | | | |
| 0.3B/8E | 1.8B | | | |
| 0.3B/16E | 3.5B | 24 | 1024 | 16 |
| 0.3B/32E | 6.8B | | | |
| 1.5B | 1.5B | | | |
| 1.5B/8E | 7.8B | | | |
| 1.5B/16E | 15.0B | 48 | 1536 | 24 |
| 1.5B/32E | 29.5B | | | |

**Table 4:** Model architectures and sizes used in our experiments. For MoE models, we replace each FFN layers with a MoE layer. $kE$ (e.g., "16E" in "0.3B/16E") represents the architecture in which each FFN layer is replaced with a MoE layer of $k$ experts. $N$: number of layers; $D$: hidden dimension of the model; $n_{\text{head}}$: number of attention heads.

## D  Model Configurations

In our experiments, we employ Lory to train decoder-only models which consists of effective parameters of 0.3B and 1.5B. For each FFN layer in the Transformer model, we replace it with MoE layers with $E$ ($E \in \{8, 16, 32\}$) experts with exactly the same architecture. Table 4 shows the configurations of model architectures.

## E  Experiments on 7B models

**Experimental Setups.** We conduct experiments on a 7B architecture. Table 5 shows the configuration of the model architectures. We train a dense 7B model and a 7B/4E MoE model. For the 7B models, we follow LLaMA2 (Touvron et al., 2023b) and use a combination of several corpora as the training set. We down-sample the full training set to a subset of 200B tokens for 7B models. Due to limited resources, we only conduct experiments on randomly batched training data for 7B models and do not apply the similarity-based batching method.

| Model | $n_{\text{params}}$ | N | D | $n_{\text{head}}$ |
|---|---|---|---|---|
| 7B | 7B | | | |
| 7B/4E | 19.7B | 32 | 4096 | 32 |

**Table 5:** Model architectures and sizes used in our 7B experiments. For MoE models, we replace each FFN layers with a MoE layer. $N$: number of layers; $D$: hidden dimension of the model; $n_{\text{head}}$: number of attention heads.

| Model | PIQA | SIQA | BoolQ | HellaSwag |
|---|---|---|---|---|
| 1.5B/8E (prompt) | **72.1** | 45.2 | **62.0** | 43.6 |
| 1.5B/8E (segment) | **72.1** | **45.6** | 60.2 | **43.9** |
| 1.5B/16E (prompt) | 71.3 | 45.0 | **56.0** | **43.7** |
| 1.5B/16E (segment) | **72.9** | **45.4** | 55.2 | 43.6 |
| **Model** | **Wino** | **NQ** | **TQA** | **Avg** |
| 1.5B/8E (prompt) | **63.7** | **7.3** | 24.2 | **45.4** |
| 1.5B/8E (segment) | 61.8 | **7.3** | **24.4** | 45.1 |
| 1.5B/16E (prompt) | 61.5 | 7.3 | **25.6** | 44.4 |
| 1.5B/16E (segment) | **62.4** | **7.6** | 25.5 | **44.7** |

**Table 6:** Downstream performance of using different inference methods. We study two routing strategy for inference. *prompt*: we make the routing decision once on the entire input prompt; *segment*: we re-route and get new merged FFNs every segment.

**Language Modeling Results.** We show the training loss curves in Figure 8 and the perplexity on held-out evaluation sets in Table 7. We find that compared to the 0.3B and 1.5B models (see Section 4), the improvement of the 7B/4E model is less significant. We think it is because (1) the similarity-based batching method is not applied in this case, making the experts under-utilized; (2) we only use four experts in the MoE model. We leave the experiments with the similarity-based batching method on MoE models with more experts as future work.

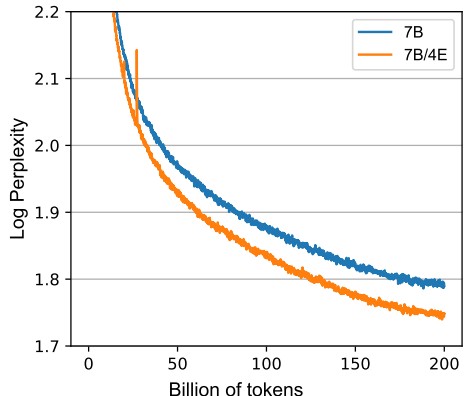

**Figure 8:** Training curves (log perplexity) of the 7B dense model and the 7B/4E MoE model. Note that when training the 7B/4E model, we do not apply the similarity-based batching method.

**Performance on Downstream Tasks.**

Table 8 shows the performance of the models on downstream tasks. We find that although the similarity-based batching method is not used when training the 7B/4E model, it still achieves clearly better results on various tasks compared to the dense 7B model. This further suggests the effectiveness of our causal routing strategy.

| Model | arXiv | Books | Wiki | C4 | Python |
|-------|-------|-------|------|-----|--------|
| 7B | 2.3 | 9.1 | 5.9 | 8.0 | 2.3 |
| 7B/4E | **2.2** | **8.7** | **5.7** | **7.7** | **2.2** |

**Table 7:** Perplexity of trained models on different evaluation sets (arXiv, Books, Wikipedia, C4, and Python). Note that when training the 7B/4E model, we do not apply the similarity-based batching method.

| | Commonsense Reasoning | | | | | Reading Comprehension | | | |
|-------|------|------|-------|----------|------------|--------|--------|-------|-------|
| Model | PIQA | SIQA | BoolQ | HellaSwag | WinoGrande | RACE-m | RACE-h | ARC-e | ARC-c |
| 7B | 76.9 | **50.2** | 65.2 | 52.6 | 66.2 | 55.3 | 40.5 | 73.0 | 38.5 |
| 7B/4E | 77.7 | 50.1 | **67.6** | **54.8** | **67.3** | **57.0** | **41.3** | **73.5** | **39.6** |

| | Closed-book QA | | Text Classification | | | | | | Avg |
|-------|------|------|--------|--------|-------|------|-------|------|------|
| Model | NQ | TQA | AGNews | Amazon | SST-2 | Yelp | Fever | MRPC | |
| 7B | 17.3 | 42.5 | 80.6 | 94.3 | 92.7 | **98.3** | 53.7 | 67.0 | 62.5 |
| 7B/4E | 18.8 | **44.5** | **81.7** | **95.7** | **93.1** | 96.7 | **57.7** | **69.7** | **63.9** |

**Table 8:** We compare the 7B/4E MoE models trained with our routing strategy (without using the similarity-based batching method) with the parameter-matched dense models on downstream tasks, including commonsense reasoning, reading comprehension, closed-book QA, and text classification.

## F  Expert Specialization: Full Results of Routing Weights

Figure 11 shows the routing weights at all layers of the 0.3B/8E model. It clearly shows that our MoE models are able to learn domain-level specialization.

## G  More Analysis and Ablation Studies

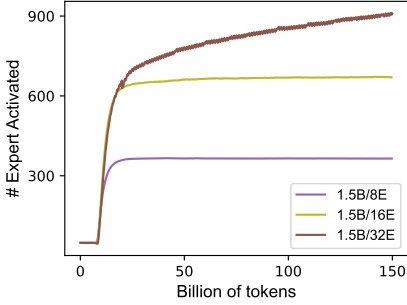

**Figure 9:** We show how many experts are actively utilized at least once every 10 training steps. We define an expert is activated if the weight is larger than $\frac{2}{E}$, where $E$ denotes the number of experts at each MoE layer. Our models have 48 layers; therefore, the 1.5B/8E, 1.5B/16E, 1.5B/32E models have 384, 768, 1536 experts in total, respectively.

### G.1  Expert Utilization

Here, we investigate the expert utilization of our MoE models. We define an expert that is activated for a given input if the routing weight is larger than $\frac{2}{E}$ (e.g., larger than 25% if there are 8 experts each layer). In Figure 9, we plot the number of experts that are activated at least once among 10 training steps (consisting of 10M tokens and 40K segments) when training 1.5B MoE models. We see that after the warmup phase at the beginning, the expert utilization quickly increases. 1.5B/8E and 1.5B/16E models have quickly utilized most of the experts; while the expert utilization of the 1.5B/32E model continues to increase until the end of the training and it is able to activate about 900 experts among 1536 experts at the end of training. This indicates that our approach is able to prevent the MoE models from

| Model | PIQA | SIQA | BoolQ | HellaSwag |
|---|---|---|---|---|
| 1.5B/8E (prompt) | **72.1** | 45.2 | **62.0** | 43.6 |
| 1.5B/8E (segment) | **72.1** | **45.6** | 60.2 | **43.9** |
| 1.5B/16E (prompt) | 71.3 | 45.0 | **56.0** | **43.7** |
| 1.5B/16E (segment) | **72.9** | **45.4** | 55.2 | 43.6 |
| **Model** | **Wino** | **NQ** | **TQA** | **Avg** |
| 1.5B/8E (prompt) | **63.7** | **7.3** | 24.2 | **45.4** |
| 1.5B/8E (segment) | 61.8 | **7.3** | **24.4** | 45.1 |
| 1.5B/16E (prompt) | 61.5 | 7.3 | **25.6** | 44.4 |
| 1.5B/16E (segment) | **62.4** | **7.6** | 25.5 | **44.7** |

**Table 9:** Downstream performance of using different inference methods. We study two routing strategy for inference. *prompt*: we make the routing decision once on the entire input prompt; *segment*: we re-route and get new merged FFNs every segment.

collapsing to dense models and achieves high expert utilization. However, when training with a large number of experts, achieving high expert utilization is more challenging.

### G.2  Inference Methods

**Comparison to segment-level routing during inference.** During inference of downstream tasks, by default, we take the task input prompt as the input of the routers in each layer and make the routing decision once. This inference method enables the decoding process to be simple and achieves low latency, since after encoding and routing the input, we do not need to use the routers again – the rest generation can be run in a (merged) dense model. As such an inference method introduces a train-test discrepancy, we study the method that routes every segment as we do during training. As shown in Table 9, routing the input once or routing each segment does not make substantial differences in the downstream tasks we evaluate. Due to simplicity and efficiency, we use the entire prompt as the routing input and perform routing only once.

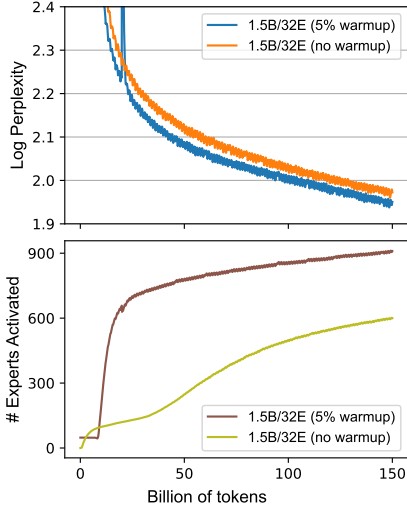

**Figure 10:** Training curves and expert utilization of employing a warmup phrase or not. We find without a warmup phrase, training leads to a worse MoE model (top) and worse expert utilization (bottom).

**Converting to sparse models for efficient inference.** In our MoE models, we merge FFN experts for each segment. This can lead to a large memory usage during inference when the model processes a large inference batch, because the parameters of a merged FFN per

batch are cached in the GPU memory. One possibility to alleviate this issue is to fine-tune the trained models with a hard-decision routing mechanism (e.g., top-k routing) after the pre-training stage. This method transitions the models with soft routers to ones with hard routers, significantly reducing the memory usage during inference. We leave further investigation on this direction as future work.

### G.3 Warmup Training

At the beginning of training (i.e., the first 5% training steps), we train a dense LM with the same configuration before training the MoE model. We initialize the MoE layers by duplicating the FFN layers of the warmup trained model. We find that this warmup phase is crucial for achieving high expert utilization especially when there is a large number of experts. Figure 10 visualizes the training loss curves and expert utilization of the 1.5B/32E model (with or without warmup training). As shown in the figure, without the warm-up phrase, the model achieves worse performance and much fewer experts are utilized.

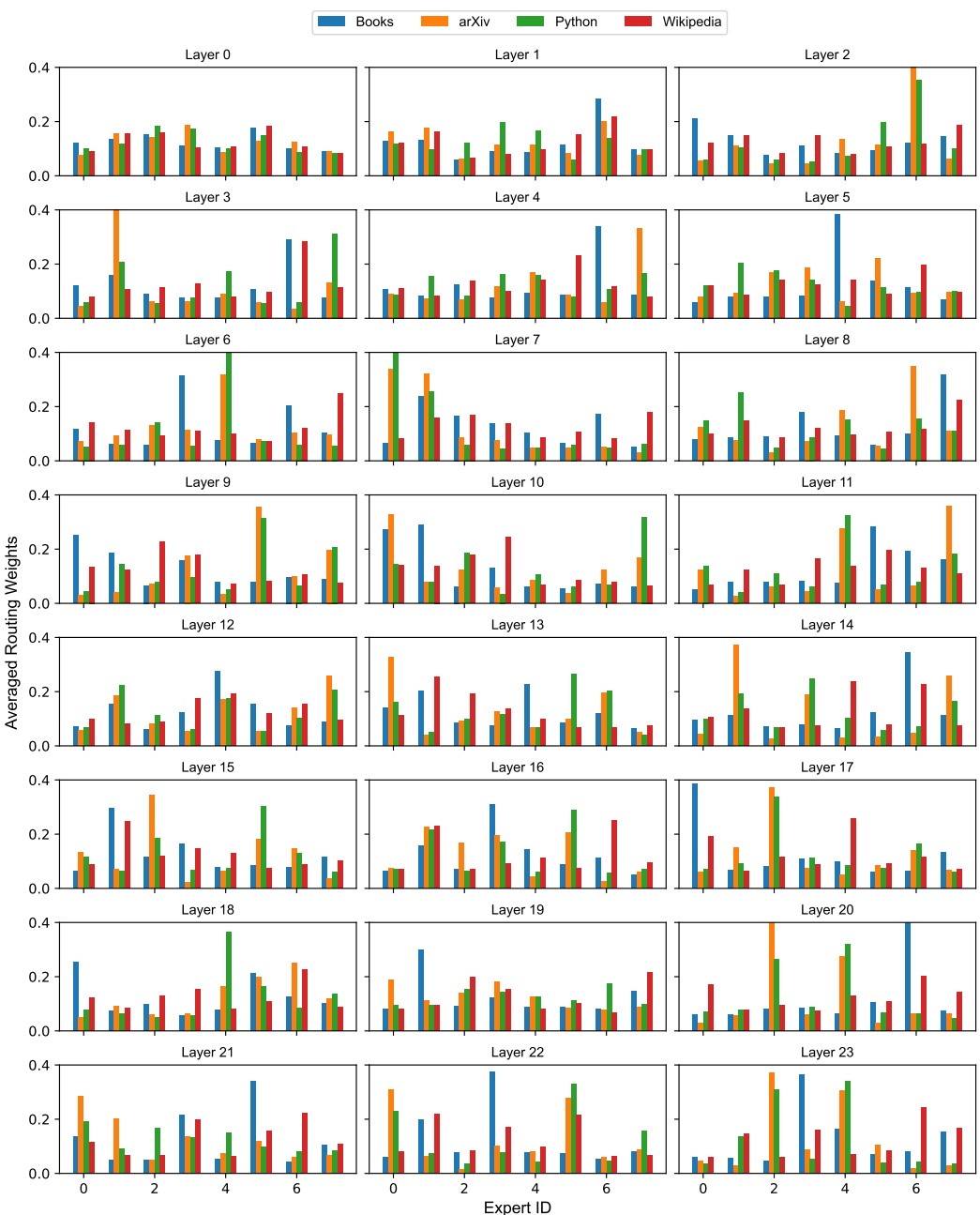

**Figure 11:** Averaged routing weights at all the layer of the 0.3B/8E model on different domains (Books, arXiv, Python, Wikipedia). We observe that the experts in our MoE models learn clear domain-level specialization.

