# OpenReview forum: "Lory: Fully Differentiable Mixture-of-Experts for Autoregressive Language Model Pre-training"
_colmweb.org/COLM/2024/Conference — COLM_

### Official Review · Reviewer_bqcN · 2024-05-03

**Rating:** 6
**Confidence:** 4
**Ethics Flag:** 1

**Summary:**

This paper makes a notable contribution towards fully-differentiable MoE LMs by proposing a couple of ways to help scale the "soft MoE" (i.e. fully-differentiable MoE) approach. Specifically, they propose:

* doing routing decisions at the segment level and not the token level. They group tokens into segments, and make subsequent routing decisions based on the segment history (some kind of averaging of hidden representations of the segments)
* a training strategy that groups semantically-similar segments, which they find leads to much better expert utilization

The claims were empirically validated at the 0.3B and 1.5B parameter-level on both perplexity (LM) tasks as well as downstream applications. Compared to equivalently-sized dense models they show improvements

**Questions To Authors:**

* "fully-differentiated" --> "fully differentiable"
* section 2.1: "exper" --> "expert"
* "We leave how to extending Soft MoE on decoder language models as the future work" --> I don't understand this - hasn't this work shown a way to extend soft MoE from SMEAR to decoder language models?

**Reasons To Accept:**

* a reasonable step towards scaling soft MoE approaches, with validation on the smaller end (<= 1.5B params), although some validation of 7B param models is done (but with no obvious improvements on downstream tasks).

**Reasons To Reject:**

* while it does seem clear that casual segment routing does better than prefix routing (and perhaps somewhat obvious?) I think the true comparison would be token-level vs. causal segment, which I believe is done in Figure 5. This Figure still shows that token-level routing is better. I'm not sure if the comparison with prefix routing really adds anything?
* section 5.3: "suggesting that a fully differentiable architecture is more effective than a sparse MoE when using the same routing strategy" - this seems a bit bold of a claim, as it's unclear if the same behaviors would extend to token-level?
* downstream improvement patterns can be explored further. For example, I noticed that for the 1.5B model the dense baseline was actually pretty good except for two specific text classification datasets (Amazon and Yelp), is there some reason why?
* the gains on 7B param models are not as large, and in fact downstream performance is mixed. This should be highlighted upfront.

---

> ### Author Rebuttal · Authors · 2024-05-31
>
> We thank the reviewer for the valuable comments and for pointing out the typos! We are encouraged that the review views our contribution as a step towards scaling soft MoE approaches! We address the reviewer’s concerns below.
>
> **Causal segment routing vs token-level routing**
>
> In general, token-level routing models have greater expressiveness compared to segment-evel routing models, because of the more fine-grained routing strategy. We did preliminary experiments with segment-level routing and token-level routing with Lory, and we observed that token-level routing does perform better. However, because token-level routing is computationally expensive, as it requires merging experts for every token, we didn’t get to run a full experiment. Instead, we compare Lory (segment routing) with EC (token routing) directly, and present results in Figure 5. We acknowledge that this will be an interesting point to show, and we’d like to add more discussions in our next revision.
>
> As for prefix routing, we consider it as a simpler alternative to causal segment routing. However,  it does not perform well. This justifies the additional considerations in Lory’s architectural design.
>
> **Downstream improvement**
>
> The performance on text classification tasks have higher variance compared to the other tasks we used (e.g., Commonsense reasoning). But we note that our 1.5B Lory models *do* clearly outperform the 1.5 dense model on all evaluation tasks (e.g., commonsense reasoning, reading comprehension).
>
> **Gains on 7B models**
>
> We are sorry about the confusion. Our 7B experiments are not complete unfortunately, as it only uses casual segment routing but not similarity-based data batching. We were not able to carry out full runs due to limited computational resources, and expect much bigger gains if we consider both techniques. We have mentioned it in the paper; we are also fine with taking out this result if the reviewer feels strongly about it. That is also the reason we left 7B results in the appendix instead of main paper.
>
> **“Hasn't this work shown a way to extend soft MoE from SMEAR to decoder language models?”**
>
> Here, Soft MoE is a particular architecture proposed in [1], and it is different from SMEAR. Soft MoE softly merges token representations instead of FFN weights. Hence, our techniques cannot be directly applied to Soft MoE for decoder model training, and we leave it as future work
>
> [1] Puigcerver et al., 2024. “From Sparse to Soft Mixtures of Experts”

---

> > ### Comment · Reviewer_bqcN · 2024-05-31
> >
> > Thanks for responding to my review. I will keep my score as is.

---

### Official Review · Reviewer_ToMC · 2024-05-10

**Rating:** 7
**Confidence:** 4
**Ethics Flag:** 1

**Summary:**

The paper looks into the problem of improving the model quality of MoE during pre-training via fully differentiable architecture. MoE models, which rely on sparsely activated experts and token routing such as top-K gating, pose training stability challenges due to their non-differentiable nature. Recent work, SMEAR, introduced softly merged experts in the parameter space, making MoE fully differentiable. However, SMEAR was primarily tested during fine-tuning for classification tasks. This paper extends SMEAR to the pre-training stage and introduces additional optimizations, including segment routing and similarity-based batching to enhance expert specialization. Evaluation shows that the proposed method can lead to lower pretraining loss than standard MoE architecture.

**Questions To Authors:**

Please discuss and compare with HashLayer.

Please provide more justification of the benefits of fully differentiable MoE vs. prior work that provide gradient estimation for discrete gating functions.

**Reasons To Accept:**

- The paper tackles an important problem, which is to improve the convergence and quality of MoE models.
- The paper builds upon SMEAR, extending its benefits from fine-tuning to pre-training. Notably, the introduced causal segment routing allows training with soft experts while maintaining the autoregressive nature of language models.
- Promising results on the tested MoE dataset and MoE architecture.

**Reasons To Reject:**

- The paper's technical novelty is limited. The core idea, e.g., the soft differentiable MoE was inherited from SMEAR, and the main difference is to extend SMEART from fine-tuning to pre-training.

- Missing discussion and comparison of related work. The paper claims that prior work used similarity-based batching only for reasoning, not expert specialization. This is not true. HashLayer, for instance, studied clustered hashes, which routes similar tokens to the same experts. The paper also got contradictory results to HashLayer without giving any explanation. The authors of HashLayer actually found that routing similar tokens to the same expert hurts the quality of MoE. More careful examination is needed before claiming similarity-based routing helps expert specialization.

- The paper does not do a thorough comparison of its approach with related work. For instance, prior work actually also addresses the differentiability of the top-K gating function, e.g, using gumble-softmax for gradient estimation and has been shown to be effective in stabilize training of MoE models (e.g., SwitchTransformer). A discussion and direct comparison of these methods would provide evidence in terms of how much benefit from a fully differentiable architecture in this work provides.

---

> ### Author Rebuttal · Authors · 2024-05-31
>
> We thank the reviewer for the valuable comments! We are glad that the reviewer found that our paper tackles an important problem and our results are promising. We address the reviewer’s concerns below.
>
> **“The paper’s technical novelty is limited”, “the main difference is to extend SMEAR from fine-tuning to pre-training”**
>
> We argue that it is important to explore new model architectures/objectives for LM pre-training in the current landscape of LLM research and development. It’s non-trivial to extend SMEAR to pre-training, due to the causal nature of generative models. To our best knowledge, we are the first to enable fully-differentiable MoE architectures for LM pre-training, which is achieved by the two techniques (causal segment routing and similarity-based data batching) proposed in our paper.
>
> **Discuss and compare with HashLayer. “The paper also got contradictory results to HashLayer without giving any explanation.”**
>
> We disagree that our results contradict HashLayer's results. HashLayer works on token-level routing, and suggests that we should route clustered tokens to different experts. Our work uses segment-level routing, which routes adjacent tokens in a segment, which are usually topically similar to the same router. Conceptually, clustered tokens do not equate tokens in the same context window. And it’s hard to make direct comparisons between the two. We will add more discussions around it in our next revision!
>
> **Benefits from a fully differentiable architecture compared to prior work on gradient estimation**
>
> We argue that the contribution of Lory is complementary - it is fully differentiable and easy to optimize. We think these solutions should be encouraged, and we hope to see future work further explore such paradigms, instead of resorting to the complicated optimization methods. In addition, gradient estimation methods still suffer from common MoE training issues such as token dropping and unbalanced loading, and are usually paired with an auxiliary loss. This complicates the training paradigm and makes it harder to tune.
>
> We note that in Section 5.3, we compare Lory with Expert Choice (EC), which is a state-of-the-art method designed to address the token dropping and unbalanced loading issues. Comparing Lory with all the other MoE architectures (e.g., gumbel softmax, switch transformer) would be computationally infeasible for us, and we’d like to defer it to future work.

---

> > ### Comment · Reviewer_ToMC · 2024-06-01
> > **Post-rebuttal comment**
> >
> > The rebuttal addressed my concerns. I raised my score from 6 to 7.

---

### Official Review · Reviewer_tkvL · 2024-05-11

**Rating:** 6
**Confidence:** 3
**Ethics Flag:** 1

**Summary:**

This paper introduces 'Lory,' a novel mixture-of-experts pretraining approach. Lory features two innovative techniques: (1) a causal segment routing strategy that efficiently merges expert contributions while preserving the autoregressive properties of language models, and (2) a similarity-based data batching method that fosters expert specialization by grouping similar documents during training. The authors pretrain a series of Lory models on 150 billion tokens from scratch, incorporating up to 32 experts and 30 billion parameters. Compared to traditional dense models, the mixture-of-expert pretraining approach demonstrates superior performance in both perplexity and in-context learning evaluation.

**Reasons To Accept:**

This paper presents a novel mixture-of-experts pre-training solution.

The experimental results demonstrate that the proposed solution outperforms traditional dense models.

**Reasons To Reject:**

The experimental setup requires improvement, particularly in terms of baseline comparison. Currently, the sole baseline is a traditional dense model. Given the availability of several off-the-shelf, pretrained, and instruction-tuned mixture-of-experts models, it would be beneficial for the authors to include these as additional baselines. Comparing the proposed solution against these models would provide a more comprehensive evaluation of its effectiveness and contextualize its performance within the current state of the art.

---

> ### Author Rebuttal · Authors · 2024-05-31
>
> We thank the reviewer for the valuable feedback. We are encouraged that the reviewer finds our pre-training solution novel and our experimental results strong!
>
> **Compare with off-the-shelf, pre-trained, and instruction-tuned MoE models**
>
> Our goal is to explore model architectures and optimization of MoE, instead of building a state-of-the-art pre-trained MoE. Our models are only trained on 150B tokens, which is far less compared to what has been used in state-of-the-art pre-trained models (trillions of tokens). In addition, instruction-tuning of MoE is orthogonal to our contributions and is out of scope of our research.
>
> In Section 5.3, we do make a comparison between Lory and Expert Choice (EC) under the same training setup. We show that Lory performs comparably to EC.

---

> > ### Comment · Reviewer_tkvL · 2024-05-31
> > **Rebuttal Acknowledgment**
> >
> > Reviewer tkvL acknowledges the rebuttal and thanks the authors for explaining why comparisons were not made to off-the-shelf, pre-trained, and instruction-tuned MoE models.
> >
> > However, the explanation does not fully address my concerns regarding the positioning of the proposed MoE solution within current MoE research.
> >
> > Therefore, I will maintain my original score.
> >
> >
> > Reviewer tkvL

---

### Official Review · Reviewer_benk · 2024-05-21

**Rating:** 6
**Confidence:** 4
**Ethics Flag:** 1

**Summary:**

1. The paper proposes a novel method to train fully differentiable MoE models for autoregressive tasks. Concretely it proposes

1.1 A segment level routing tasks that computes the expert mixing weights for a segment based on the (aggregate) hidden representation from the previous segment, and uses that to softly mix the experts in parameter space

1.2 Leveraging similarity based data batching, similar to In-context pretraining [1] to avoid expert under-utilization, that subsequently results in better domain specialization of experts

2. The authors demonstrate superior performance on both perplexity as well as downstream tasks compared to an (active) parameter matched dense model setup across a series of compute budgets. They also demonstrate competitive performance compared to token level hard-routing baselines (Expert Choice [2])

3. Furthermore, they also demonstrate that the proposed use of similarity based batching benefits MoE training more than it's random sampling counterpart, and that strong performance gains are observed even for dense models (with the dense sim-batch model outperforming the MoE random batch baseline).

4. The paper also demonstrates that the proposed method achieves good domain level expert specialization without any domain level supervision.

[1] Shi, Weijia, et al. "In-Context Pretraining: Language Modeling Beyond Document Boundaries." arXiv preprint arXiv:2310.10638 (2023).

[2] Zhou, Yanqi, et al. "Mixture-of-experts with expert choice routing." Advances in Neural Information Processing Systems 35 (2022): 7103-7114.

**Questions To Authors:**

1. The expert choice with segment level routing model (EC, segment-level in Figure 5) is barely better than the dense model (Figure 3). What is the intuition on the poor performance ?

2. For the first segment routing, why not use a BOS segment of 1 token length, and use that for priming the mixture weights for all subsequent generations ? Otherwise, you might be limiting the model from learning from the first segment. For example, if the model used absolute position embeddings, then if I understand correctly, the embeddings for the first segment would never get trained.

2. [Minor] It would be good to have the notation for e_{i} to be consistent between Equation (1), (2) and (3). Maybe consider changing Eqn (1) to be Top-k over (e_i) ?

**Reasons To Accept:**

1. The proposed method allows the previously proposed SMEAR approach to be used for autoregressive modeling with the segment based routing approach, potentially allowing for fully differentiable MoE models to be scaled.

2. The discussions on parallelism, especially Appendix H is very informative, and in my opinion forms for a good discussion point for scaling fully differentiable MoE models compared to the hard decision based token routed MoE models

3. The authors demonstrate the utility of ICL pre-training for MoE model training, showing improved gains compared to random batching. While the results are preliminary, that forms for a good avenue of exploration for training MoE models

**Reasons To Reject:**

My concerns with the paper are as follows:

1. A fundamental mismatch between training and test objective:
While the paper does discuss the train-test mismatch in Appendix G3, one thing to note is that for most of the downstream tasks, the distribution between the prompt-lengths and the generation lengths is not very varied. Furthermore, without an ablation on the segment length, it's hard to understand at what point the segment length is too large for the routing function to be learned properly. In my opinion, some experiments to test this hypothesis are important. For example:

1.1 Controlling for prompt-length: By averaging the hidden representations for generating the FFN weights, intuitively there would be a diffusion of information. If that is indeed the case, then a generative needle in a haystack ([1]) or a multi-key NIAH ([2]) would be interesting to test if this diffusion indeed causes issues in learning the routing weights

1.2 Another controlled experiment that can help measure this is via perplexity measures: specifically, by fixing a prompt length and completion length to a fixed value, and then measuring the perplexity on the completion as a function of the segment length for a trained model. The difference in perplexity should be a good indication on how much impact does the train-test mismatch actually have.

1.3 Another test that is important to understand the (potential) limitations for this approach would be to have generative tasks that are long (i.e the generations are >> segment length). Intuitively, with a larger generation, the model should progressively adapt to use different experts based on the prompt + generation content, but that would not happen based on the proposed inference methodology. It would be good to quantify this limitation if possible.

2. Comparison with token level MoE routing:
Based on the experiments presented in the paper, it is hard to understand what advantages would having the soft MoE approach have compared to vanilla token based routing.

2.1 If the hypothesis is that token based routing requires advanced training techniques like Expert Choice to be better than the proposed method, it would be good to demonstrate the proposed methods superiority compared to a vanilla top-k based routing approach

2.2 If the hypothesis is that the proposed routing achieves better OOD generalization because of expert specialization, it would be good to demonstrate that. Concretely, in the current paper, it is hard to disentangle if the expert specialization comes from the sim-batch data approach or from the proposed routing (or rather, if the token routed models were also trained in the sim-batch setup, would they also result in expert specialization).

[1] Kamradt, G. "Needle in a Haystack–pressure testing LLMs." (2023).

[2] Hsieh, Cheng-Ping, et al. "RULER: What's the Real Context Size of Your Long-Context Language Models?." arXiv preprint arXiv:2404.06654 (2024).

---

> ### Author Rebuttal · Authors · 2024-05-31
>
> We thank the reviewer for the valuable comments! We are glad that the reviewer sees great potential in training large-scale fully-differentiable MoEs. We address the concerns below.
>
> **Mismatch b/w training and testing**
>
> Thanks for the excellent suggestions! In our paper, we showed that a simple inference strategy works well for a wide range of tasks, for both classification and generation. We agree that the inference strategy requires further investigation for long-form generation. However, a comprehensive study demands careful designs and evaluation, and due to computational constraints, we’re unable to delve deeper into this during rebuttal. We’ll explore these suggestions in future work.
>
> **“Expert specialization comes from the sim-batch data approach or from the proposed routing”?**
>
> In Appendix G.1, we compare Lory and EC, both trained with sim-based batching. We show that Lory is better at modeling OOD areas such as coding than EC. This should sufficiently show that the specialization does not fully come from the batching strategy, but also due to Lory’s architectural designs. We didn’t do a full comparison on Lory between with or without sim-based batching, and we’ll add discussions around it.
>
> **Comparison with token level MoE**
>
> In Figure 5, we show Lory performs similarly to EC. We’d have liked to compare our method with more approaches, e.g., gumbel-softmax. However, due to computational limitations, we only got to experiment with one baseline and opted for EC, which has shown to be the SoTA MoE architecture.
>
> **Poor performance of segment-level EC**
>
> Our hypothesis is that EC is less effective in segment-level routing. EC can lead to the token-dropping issue; while in segment-level EC, it may drop the whole segment completely, which may even more severely hurt the performance.
>
> **“The first segment would never get trained”**
>
> Sorry about any confusion, but this is not true. The stop-grad is only applied on the merged FFN for the first segment, to avoid gradients from the first segment to update the FFNs and routers, thereby avoiding information leakage. However, the first segment will still be used to update the attention layers, embedding layers, etc.
>
> **“Why not use a BOS segment?”**
>
> Good point! We indeed conducted a similar experiment, where we make the first segment’s routing weights as trainable parameters. We find that doing so leads to slightly worse performance.
>
> **Notion for e_{i}**
>
> We will revise Eq (1) to make it consistent.

---

### Decision · Program_Chairs · 2024-07-10

**Decision:**

Accept

**Comment:**

Lory is a fully-differentiable MoE architecture, designed for autoregressive language model training. The reviewers agree to accept the submission. It's a good exploration given the context of large-scale MoE systems for LM training. The reviewers ask to include more discussions about limitations and controlled experiments.